# Population Analysis Identifies 15 Multi-Variant *Dominant White* Haplotypes in Horses

**DOI:** 10.3390/ani14030517

**Published:** 2024-02-05

**Authors:** Aiden McFadden, Micaela Vierra, Holly Robilliard, Katie Martin, Samantha A. Brooks, Robin E. Everts, Christa Lafayette

**Affiliations:** 1Etalon Inc., Menlo Park, CA 94025, USA; mvierra@etalondx.com (M.V.); hrobilliard@etalondx.com (H.R.); khoefs@etalondx.com (K.M.); reverts@etalondx.com (R.E.E.);; 2Department of Animal Sciences, UF Genetics Institute, University of Florida, Gainesville, FL 32611, USA; samantha.brooks@ufl.edu

**Keywords:** depigmentation, *KIT*, *Dominant White*, horse, white spotting

## Abstract

**Simple Summary:**

Mutations in the *KIT* gene cause many different types of white spotting in the coats of domestic horses, similar to other mammals. Although there are 36 different alleles responsible for causing different phenotypes, there have been no comprehensive observations performed on horses with haplotypes of two or more *KIT* variants. Because some *KIT* alleles are likely lethal in the homozygous state, some believe that it is impossible for a horse to carry three or more white spotting alleles. Here, we identify 15 unique haplotypes with two or more mutant *KIT* alleles and present data on a subset of their phenotypes. Identifying haplotypes helps to better understand the mechanisms responsible for white spotting in horses and allows for better control of breeding practices. We sought to prove that horses can carry more than two functionally annotated *KIT* variants and document any differences in white spotting compared to horses with only a single variant.

**Abstract:**

The influence of a horse’s appearance on health, sentimental and monetary value has driven the desire to understand the etiology of coat color. White markings on the coat define inclusion for multiple horse breeds, but they may disqualify a horse from registration in other breeds. In domesticated horses (*Equus caballus*), 35 *KIT* alleles are associated with or cause depigmentation and white spotting. It is a common misconception among the general public that a horse can possess only two *KIT* variants. To correct this misconception, we used BEAGLE 5.4-phased NGS data to identify 15 haplotypes possessing two or more *KIT* variants previously associated with depigmentation phenotypes. We sourced photos for 161 horses comprising 12 compound genotypes with three or more *KIT* variants and employed a standardized method to grade depigmentation, yielding average white scores for each unique compound genotype. We found that 7 of the 12 multi-variant haplotypes resulted in significantly more depigmentation relative to the single-variant haplotypes (ANOVA). It is clear horses can possess more than two *KIT* variants, and future work aims to document phenotypic variations for each compound genotype.

## 1. Introduction

The domestic horse is prized for its diversity of coat colors ranging from black to white with many shades in between. The desire to breed horses of specific colors has led to efforts to understand the genetics of equine coat color. The etiology of melanogenesis is complex, as multiple genes influence pigment production, altering base coat color, while others cause variations in the distribution of melanocytes, yielding a variety of markings. Horses with identical genotypes can vary in color and pattern as a result of random events during development, and horses with different genotypes can manifest phenotypes so alike as to be indistinguishable to the eye, highlighting the complexities of gene interactions within pigmentation pathways.

One gene central to mammalian pigmentation phenotypes is the proto-oncogene Receptor Tyrosine Kinase (*KIT*) on ECA3 [1,2]. *KIT* encodes a transmembrane signaling protein involved in melanogenesis, hematopoiesis, stem cell maintenance and gametogenesis. During melanogenesis, melanoblasts (precursors to pigment-producing cells) derived from the neural crest migrate to populate skin, hair and other tissues. The KIT protein is responsible for transmitting signals during melanogenesis required to properly express pigment-related genes. The KIT protein binds with the KIT ligand in its extracellular domain, causing self-dimerization and subsequent kinase activity within the intracellular domain. For melanogenesis, the KIT protein phosphorylates Melanocyte Inducing Transcription Factor (MITF), which upregulates pigment-related genes in its phosphorylated state [1,2]. Loss-of-function *KIT* mutations result in hypopigmentation in mice, humans and horses, likely due to interruptions in the migratory signaling [2,3,4,5,6]. Mutations can also affect other KIT protein pathways, leading to embryonic death, sterility, growth issues and immunodeficiencies [5].

Over 35 mutations associated with or causing depigmentation were previously identified within equine *KIT* [3,4,5,6,7,8,9,10,11,12,13,14]. These include *Sabino 1* (*SB1*) and the *Dominant White* series of alleles (*W1–W35*) which are a combination of coding and non-coding mutations within *KIT*. Many *Dominant White* variants arose recently and have known founders [4]. Most of these alleles are predicted to affect the expression of the wild-type protein or produce a less active KIT protein and are inherited in an autosomal dominant or incomplete dominant manner [3,4,5,6,7,8,9,10,11,12,13,14]. The *Tobiano* (*TO*) allele is a large paracentric inversion outside of *KIT* thought to interrupt its regulation, resulting in large white areas on the coat [4]. Individuals of diverse breeds including Paint Horses and Gypsy Cobs were observed carrying both the *TO* and *W20* or *SB1* alleles [15]. *Roan* is a white spotting trait also mapped to equine *KIT* [16,17]. Although there are multiple mutations associated with *Roan* in various breeds, the causative mutation is unknown [16,17]. Many *KIT* mutations are thought to be homozygous embryonic lethal (e.g., *W22*, *Roan*) due to a lack of observed homozygotes and an observed Mendelian ratio of 2:1, but other mutations (e.g., *SB1*) are viable and further decrease pigmentation in the homozygous state [4,18]. Some data have supported the viability of homozygous *Roan* horses [16].

Although specific combinations of *KIT* variants are predicted to be homozygous embryonic lethal, there are no data suggesting that viability is limited to horses with two or fewer *KIT* mutations. As the first *Dominant White* mutations were suspected to be homozygous lethal, it is a common misconception in the horse community that a horse may only have up to two *KIT* variants even though multiple studies have identified variants in complete linkage or individuals having three *W* alleles [3,8]. Although each horse possesses only two copies of *KIT*, it can possess multiple non-reference alleles on the same chromosome. If the sister chromosome also possesses a *KIT* variant, then an individual may possess three or more *KIT* variants. Our goal was to perform a retrospective study of horses in our database to understand the number of *KIT* variants that could be present in a single horse to develop a broader understanding of which *KIT* variants can be found in linkage with each other and thus define *KIT* haplotypes and the resulting phenotypes. We hypothesized more deleterious variants would not be found in the homozygous state or in a compound heterozygous state with other deleterious variants as they exist in low frequency and may not be tolerated. We expect more tolerated variants to be found in complex haplotypes with other variants of the same nature as they exist in higher frequencies [15]. Here we report 15 unique haplotypes with two or more *KIT* variants found in a sampling of 8334 horses.

## 2. Materials and Methods

### 2.1. Horses

All obtained horse samples followed the International Guiding Principles For Biomedical Research Involving Animals. No experiments were performed on the horses as hair samples were provided by willing horse owners with informed consent. From the horses submitted to Etalon, Inc., for commercial genetic testing, a subset of 8334 horses, representing diverse lineages including Stock Horses, Thoroughbreds, Warmbloods, Iberians, Carriage Horses, Heavy Horses, Arabians and Ponies, were considered for our haplotyping study. Genomic DNA was extracted from hair samples or retrieved from previously extracted gDNA, where applicable, and sequenced on the commercially available platform of Etalon, Inc., with 2 × 150 bp reads on a NextSeq1000 [3]. Reads were filtered and aligned to EquCab3.0 following best practice guidelines [19]. For examination of the relationships between haplotypes and spotting phenotypes, quality photos displaying the full body of each horse, including face and legs, were used for the phenotyping of each horse (Figure 1). A total of 321 horses bearing at least one *Dominant White* variant, and for which a photo could be obtained for phenotyping, were used for the phenotype study.

### 2.2. Variant Screening

VCF files were generated for each horse using sequencing data from Etalon’s commercially available platform using best practice guidelines [19]. Each horse was genotyped for all 36 documented *KIT* (NM_001163866.1) variants between the terminal ends of the untranslated regions and previously associated with depigmentation (Table 1). Horses were sorted into groups by the variant and number of copies they possessed. Horses with one variant comprise the Single groups and include the *SB1*, *W5*, *W15*, *W19*, *W20*, *W22*, *W32*, *W34* and *W35* alleles. Horses in the Single groups were also separated by zygosity (e.g., *W20/n* is a different group than *W20/W20*). We filtered and sorted the Compound group for unique combinations of three or more variants (e.g., *W20/W20 W35/n* vs. *W20/W20 W35/W35*).

### 2.3. Phenotyping and Statistical Analysis

A single blinded observer graded depigmentation from a single photo of each horse according to a system reported in a previous study [3] and adapted from the work of Rieder et al. [6]. The qualifications of the single observer were as follows: familiar with horses and coat color phenotypes, genetics background and basic understanding of *Dominant White* variants. They were provided with an instruction sheet and Appendix A to aid in phenotyping. In the original method, which was primarily focused on facial and limb depigmentation, the expected phenotype of horses with *KIT* mutations was based on comparative phenotypes in rodents. The scoring system was adjusted to give more detailed consideration to the abdomen and hindquarters, which are commonly depigmented by spotting variants in horses [4]. The modified system utilizes a scale of 0 to 38 depending on the extent of white markings, where 38 represents a fully white horse with two blue eyes (Appendix A). ANOVA was used to reveal associations between average white scores (AWSs) and haplotype combinations.

### 2.4. Haplotype Identification and Frequency Calculations

As the variants span all of *KIT* (~80 kb from first to last exon), directly observing the phased haplotypes with long-read sequencing was impractical for all samples in this large dataset. To overcome this obstacle, we phased VCF files using BEAGLE 5.4 but performed no imputation on genotypes [20]. Only horses with complete genotyping for all 36 *KIT* variants were considered. Allele frequencies were calculated from the genotypes of 8334 horses using the vcftools–freq command [21]. We created a custom Python V3.11 script to analyze the results of BEAGLE phasing. Phased VCF files were collected into a single dataframe and queried for unique haplotypes and haplotype frequency using pandas V2.0.1 [22].

## 3. Results

All 36 markers for 8334 horses were successfully phased by BEAGLE with an estimated population size of 152,922 and an estimated error of 2.5 × 10^−6^ as calculated by BEAGLE. Allele frequencies and white scores of each marker under consideration are shown in Table 2. We found that 2891 horses (34.68%) in our sample had one *KIT* variant associated with depigmentation, 1764 horses (21.16%) had two variants, 364 horses (4.36%) had three, 66 horses (0.79%) had four, 3 individuals had five variants and 1 individual had six variants. Not every *KIT* variant was present in our sample set, as these variants are limited to individual families not represented in this population [*W1*, *W2*, *W4*, *W6–W12*, *W14*, *W16–18*, *W21*, *W23–W28*, *W33*] [4]. We identified 72 multilocus genotypes comprising 28 haplotypes (Appendix A). The frequency of each haplotype in our sample is shown in Table 3. We sourced photos for 161 horses with three or more variants, and all of these 161 horses were used in our phenotyping study along with 160 horses selected as controls for possessing only one variant with regard to zygosity.

We identified 10 haplotypes with two *KIT* variants and 5 haplotypes with three *KIT* variants. Consistent with previous reports, the *W22* allele was found in complete linkage with the *W20* variant [8,15]. We identified the *W19* allele in linkage with both the *W34* and *W35* alleles and by itself. Previous studies reported *W19* to be linked to *W34*, but these individuals also carry a copy of *W35* in phase [3]. Homozygosity was observed for wild-type, *W15*, *W19*, *W20*, *W32*, *W34*, *W35* and *SB1* alleles. The *W32SB1*, *W32W20*, *W32W35*, *W34W35* and *W19W34W35* haplotypes were also identified in the homozygous state. The most common alleles in our dataset were *W35* (16.8%) and *W20* (20.5%). Multi-variant haplotypes had a frequency of 7.1% in our sample with *W32W35* being the most common haplotype (3.32%).

We successfully sourced at least three photographs for 12 of the identified multilocus genotypes (Table 4) and for each of the corresponding 13 alleles with only one variant (Table 2). Eight multi-variant haplotypes were found to cause significantly more depigmentation relative to their corresponding single-variant groups. Four of the eight multilocus genotypes had an AWS greater than 20, representing extensive depigmentation of the face and legs. *SB1W32/W32W35* had the highest average depigmentation of the analyzed compound genotypes with an AWS of 32.66 (*n* = 3). Four of the twelve multi-variant haplotypes were found to not cause significantly more depigmentation than the variant-matched Single groups. *W32/W32W35* had the lowest AWS of all compound genotypes (4.66). The AWS of individuals with only the *W32* allele was 2.63, the lowest of all variants investigated in our study. *W3*, *W5*, *W15*, *W19*, *W30*, *W31* and *SB1* had AWSs greater than 20, and these alleles, excluding *W15* and *W19*, typically resulted in an all-white horse.

There were many multi-variant genotypes we were not able to include in our phenotyping study, because of a small sample size. Twelve multi-variant genotypes with one or two phenotyped horses were not included in our statistical analysis due to sample size, but we report white scores for these as notable cases as they represent previously unreported viable combinations of *KIT* variants (Appendix A). One of these twelve multi-variant genotypes is *W20/W20W22*, which was previously reported to cause an all-white phenotype [8,15]. The individual identified with this genotype in our dataset also had an all-white phenotype with a white score of 36.

## 4. Discussion

The current common practice for reporting *KIT* variants in commercial testing reports (Variant1/Variant2, e.g., *W32/W34*) is limited to two variants and assumes those variants are on opposing copies, which is not always the case. To date, *W22* and *W19* are observed in linkage with other *Dominant White* alleles [3,8,15]. Our data support documented linkages as every *W22* allele was found with at least one *W20* allele and most *W19* alleles were found with one *W34* and one *W35*. It is understood that *W22* and *W20* are linked, and it has been reported that *W19* is linked to *W34* [3,8,15], but this is the first report that *W19* is in partial linkage with *W34* and *W35*. Although some alleles are known to be linked, variants are currently reported to horse owners as *W*^X^/*W*^Y^, where X and Y denote different *KIT* loci regardless of phasing (e.g., *W20/W22* or *W22/n*). Reporting a separate genotype for a linked allele like *W22/n* fails to provide information on the true haplotype. From this notation, one would conclude only the *W22* variant is present, but to date, the *W22* variant has only been observed on the *W20* background [8,15]. More problems arise when individuals have three or more variants like a *W20/W20W22* or *W19W34W35/n* genotype. Reporting genotypes as *W20/W22* groups two loci together, assumes they are out of phase and does not provide information about the other loci. This report could be taken to mean either *W22/n W20/n* or *W22/n W20/W20* because it reports information for only two of the four alleles. Although *W22* likely originated on the *W20* background, *W22* may not always remain linked to *W20* [8,15], and including the in-phase *W20* allele as part of the *W22* allele shows an incomplete compound genotype. *Dominant White 19* is partially linked to both the *W34* and *W35* variants, so reporting only *W19* fails to inform horse owners whether a horse possesses the linked alleles or not. Reporting diverse alleles as one locus misrepresents these genotypes by implying that only two variants are possible. However, reporting the entire compound genotype for *KIT* variants ensures that information is accurately communicated and allows for the future possibility that a recombination event could separate or join alleles.

Reporting of *Dominant White* variants does not intuitively distinguish between the viability or phenotypes of different variants. Multiple *W* variants are reported to be viable and cause more depigmentation in the homozygous state relative to a heterozygote (reviewed in [4]). Therefore, the name “Lethal White” or “Lethal Dominant White” has become a misnomer since some variants found in the homozygous state cause incompletely dominant and viable phenotypes. Designating the variants including but not limited to *W20*, *W32*, *W34* and *W35* as members of the *Sabino* group would have been more appropriate, given their phenotype and pattern of inheritance. Some research more appropriately refers to the *W* locus as *White Spotting* instead of *Dominant White* [3,9,14], in which case there is an arbitrary distinction between the *Sabino 1* allele and *KIT* variants causing sabino-like phenotypes. Variants not yet observed in the homozygous state may be lethal when in combination with other variants of the same nature (reviewed in [4]), although for rare alleles there may not yet have been enough crosses of carriers to determine this conclusively. To date, there is no clear and accurate resource for inferring the possibility of a homozygous lethal effect without reading each *Dominant White* publication. Information regarding which combinations of *Dominant White* variants are lethal is limited in quantity and hard to source. This complicates breeding practices as owners are unsure which combinations could result in reduced fertility rates.

To solve both of these major inaccuracies, variant reporting in commercial testing should have the flexibility to indicate (1) more than two variants, (2) the full genotype of each variant and (3) a method for distinguishing which *Dominant White* variants are likely lethal. Following these guidelines to convey more accurate and intuitive results would help horse breeders better understand their animals and ensure genotyping information is not omitted in scientific reports. Reporting each variant as its own genotype (e.g., *W32/n W35/n SB1/n*) would result in all genotype information being included. Alternatively, when the phase is known, one can report phased data such as *SB1/W32W35* to show which mutations are on the same strand. Indicating potential lethal homozygous variants in commercial reports would help customers better understand the data they are receiving. Breeders would be able to distinguish between variants and make decisions to better avoid combinations likely to lower fertility rates. Reporting this extra information would allow for more accurate science and ensure lethal combinations are easier to avoid.

A common assumption in the equine community is that an individual can only carry up to two *KIT* mutations because of the hypothesized lethality of homozygosity and outdated scientific information [5]. From 8334 horses analyzed, we produced 16,668 haplotypes. Of these, 1191 contained two or more mutations. Overall, there were 434 horses that had three or more mutations, and the most we saw was six in a horse with *W19/W19*, *W34*/*W34*, *W35*/*W35*, which showed *W19W34W35* to be a three-mutation haplotype. Although haplotypes with multiple functionally annotated mutations are rare, they are present in horse populations. The most common haplotype was *W32W35* (3.31%) and was identified in the homozygous state. Because this genotype comprises the second (*W35*) and third (*W32*) most common alleles, the chances of producing homozygous individuals are much higher. The *SB1*, *W32* and *W34* alleles were more commonly found in haplotypes than by themselves. We only identified six *SB1* alleles and eight *W34* alleles out of phase with other variants but observed hundreds of haplotypes including *SB1* or *W34*. *W32* was more commonly identified in phase with other *KIT* variants than it was identified alone. There are likely other combinations of white alleles not present in our dataset as multiple alleles are found only in certain family lines and these familial white alleles might have occurred on the background of other *KIT* alleles similarly to *W22* [3,8,12,15]. Similar to initial reports, *KIT* alleles *W3*, *W5*, *W13*, *W22*, *W30* and *W31* were not found in homozygous states (reviewed in [4]). These alleles are possibly lethal in the homozygous state, but due to their low frequency, that remains to be confirmed. It is likely that deleterious variant combinations, such as frameshift and stop-gain mutations, are non-viable in the homozygous state or in combinations with other deleterious alleles. Haplotypes containing the *SB1*, *W19*, *W20*, *W32*, *W34* and/or *W35* alleles were viable in the homozygous state, and many combinations of these variants exist in our sample. We did not study the health of each horse with three or more *KIT* variants; future studies focused on the health of horses with more than two *KIT* variants are needed.

There is a large distribution of white scores for the *Dominant White* alleles reflecting the differences in toleration. *W32* is likely the most tolerated variant with the least effect on the KIT protein as it has the lowest AWS and many individuals with only *W32* are solid (non-white, minimally white) in color (reviewed in [4]). It is possible that the *W32* variant is linked to an undiscovered white-causing mutation in certain breeds (reviewed in [4]). Consistent with previous reports, *W20*, *W32*, *W34* and *W35* were found in many breeds [4,15], and these variants cause mild white spotting or white boosting phenotypes when in or out of phase with other variants. *Dominant White* variants *W1-W19*, *W21-W28*, *W30* and *W31* produce phenotypes with depigmentation over 50% and are likely less tolerated than variants producing mild sabino-type phenotypes (reviewed in [4]). Variants *W3*, *W5*, *W13*, *W30* and *W31* are likely the least tolerated variants found in our sample as they have not been observed in the homozygous state, are predicted to have extreme functional impact (reviewed in [4]) and were found to have AWSs over 25 in our analysis. The combination of deleterious alleles such as *W22* with tolerated alleles like *W20* out of phase results in higher white scores [8,15]. Although we cannot conclude the effects of unphenotyped haplotypes, it is likely that combining multiple white alleles results in less active KIT protein, magnifying the extent of white markings similar to the effect *W20* has on the *W22W20* haplotype [8,15].

Having a white variant does not mean a horse must have white markings, simply that it is more likely to develop markings during melanogenesis. Solid-colored horses were observed for white-causing mutations such as *W20*, *W32*, *W34* and *W35*. We identified multiple solid-colored horses with the *W20/W32W35* and *W35/W34W35* genotypes. The *Tobiano* and *W35* alleles have been reported to not always produce any white markings (reviewed in [4,23]). The amount of depigmentation is highly variable for single alleles and allelic combinations of these variants, even within the same genotype, because of the stochastic nature of pigmentation (Figure 2). Some registries will reject horses based on white markings, but because multiple white variants do not always produce white markings, members of the same family do not always qualify for registration in the same studbooks. This highlights the importance of genetic testing and using genetics as the basis for in/exclusion instead of phenotypes.

There were many allelic combinations that we were not able to test for significance as we could not source more than two photographs of individuals possessing the compound genotype. However, based on the photographs, all ten compound genotypes with *n* < 3 showed more depigmentation relative to the variant-matched Single groups. We predict, therefore, more depigmentation on average for all allelic combinations where we observed one or two individuals because of the large increases in white scores. If one of the alleles is a high-impact variant (frameshift, indel, stop-gain, active residue mutations), then additional moderate variants (*W20*, *W32*, *W34*, *W35*) are likely to produce more divergent phenotypes. However, in the absence of high-impact variants, allelic combinations are likely to produce sabino-like phenotypes with varied amounts of white markings.

There were a number of limitations present in our study: (1) The largest limitation is the sample size of photographic records compared to genetic records. The lack of photographs makes it harder to accurately predict the amount of white spotting that will occur for each given compound genotype. In-person phenotyping would improve limitations to photographs as the entire horse would always be scored, but it is complicated by the geographic diversity of this population. However, the benefit of in-person phenotyping would likely only impact white scores by 1–2 points in the case where white markings are only present on one side of the body. (2) Since some variants are breed-specific, allele frequencies calculated in our sample might be skewed due to an unequal sampling of breeds. We believe this skew is minor for more common variants as our frequency of *W20* is similar to that in a previous report [15]. The skew for rarer variants might be larger in magnitude as their low frequencies increase sampling bias. Since no variants thought to be rare were found with high allele frequencies (>0.5), we believe this skew to be minor. (3) Rare variants might be incorrectly phased by BEAGLE because of low allele frequencies. *W13* was identified both in and out of phase with *W35*, but only two copies of *W13* were identified in our data. We cannot be sure if both results are accurate or if one was phased incorrectly.

Resequencing or reanalyzing horse populations with rare variants would help to reveal possible linkages between older and recently annotated white spotting variants. Our grading system favors markings on the legs and face, with minor consideration of the body, making it difficult to employ the same strategy to evaluate interactions with the *Tobiano* allele, which produces large white patches on the body [23]. Future work may need to alter our phenotyping strategy or use another method to give more consideration to these areas. Despite these limitations, it is clear individuals with more than two functionally annotated *KIT* variants are viable.

## 5. Conclusions

We screened a database of 8334 horses to collect genotypes for *KIT* mutations implicated in depigmentation. After phasing this dataset using BEAGLE, we identified 434 horses possessing three or more *KIT* variants with 72 unique multi-variant genotypes. These 72 combinations comprise 28 unique haplotypes where 1 is the wild-type, 12 haplotypes have one variant, 10 have two mutations, and 5 haplotypes have three. We sourced photos for 161 individuals for 12 compound genotypes and 160 horses with only one *KIT* variant and quantified depigmentation using a method reported in previous white association studies. We found seven compound genotypes that result in more depigmentation than the single-mutation counterparts and five combinations were not found to increase relative depigmentation. We provided the white scores for 12 additional cases of white spotting for 10 other multilocus genotypes. In general, combining variants with major protein impact (fs, indels, stop-gain variants) with other mutations increases the chance of extensive depigmentation. Variants with minor impact (*W20*, *W32*, *W34*, *W35*) typically result in similar amounts of depigmentation in tandem but are highly variable. We propose *KIT* variants be reported as phased genotypes or the separation of different loci to more accurately report genotypes.

## Figures and Tables

**Figure 1 animals-14-00517-f001:**
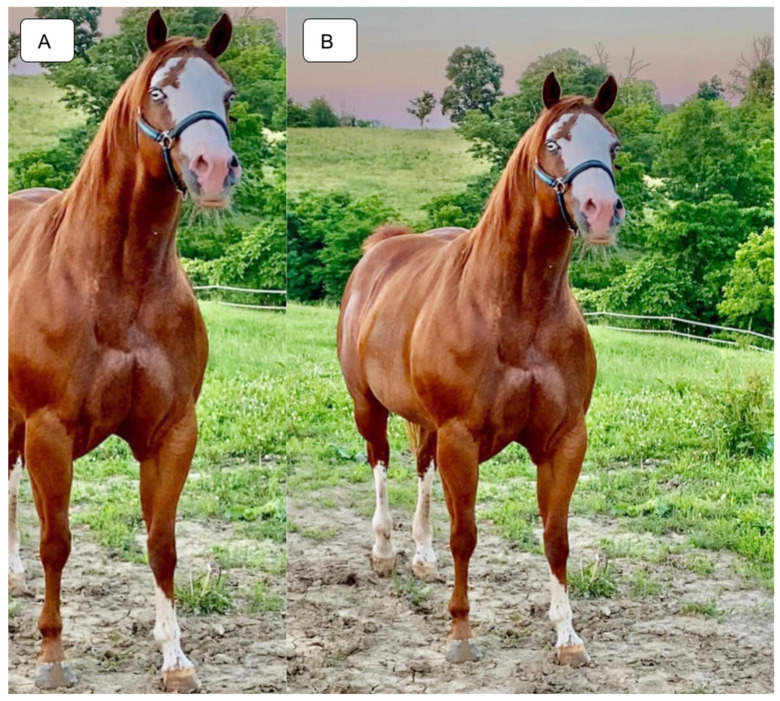
Exclusion criteria for phenotyping study. The same image is shown in both panels, but only Panel (**B**) is valid for the phenotyping study. The photograph in (**A**) does not show the hind quarter and cannot be accurately scored because of the missing limbs. Figure (**B**) shows the entire horse, but information on the other side is still lost. However, the information lost is minimal as only minor consideration is given to the body.

**Figure 2 animals-14-00517-f002:**
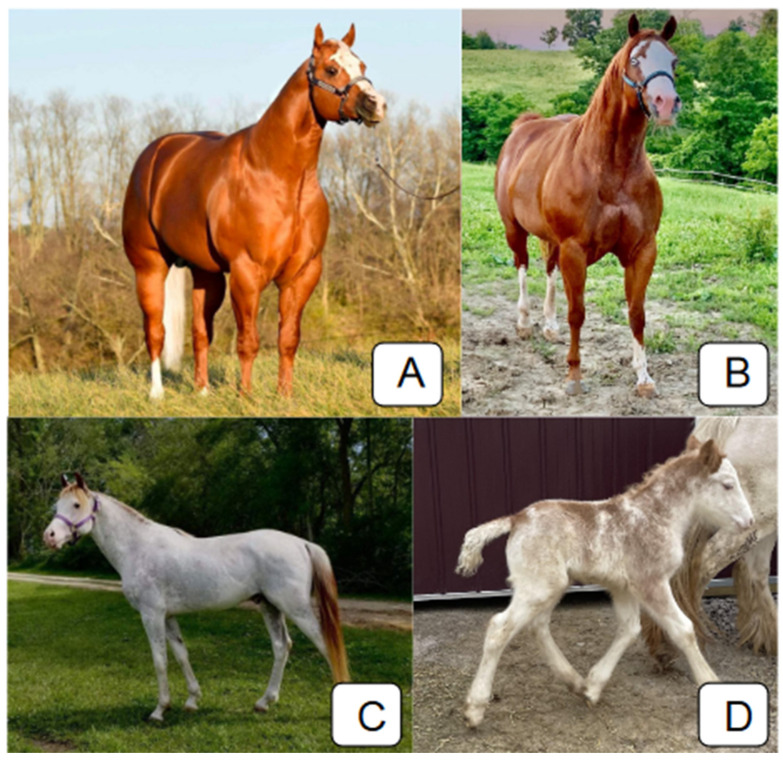
Phenotypes of various *KIT* genotypes. The horses in (**A**,**B**) have identical genotypes for white variants (*W35*/*W34W35*) but display differences in depigmentation. The horse in (**B**) has two depigmented eyes, and it has broader white marks on the face and more depigmentation on the legs than the horse in (**A**). The horse in (**C**) has five white variants (*W34W35/W19W34W35*) and is maximally depigmented but has non-blue eyes. The horse in (**D**) has three white variants (*W20/SB1W32*) and is over 50% white. All photographs in this image were also used for phenotyping.

**Table 1 animals-14-00517-t001:** All *KIT* variants associated with depigmentation with the type of mutation and location on the EquCab3.0 genome. Adapted from [4].

Allele	Coordinates	Type	Phenotype
*W1*	chr3:79545942G>C	nonsense	All White
*W2*	chr3:79549540C>T	missense	All White
*W3*	chr3:79578535T>A	nonsense	All White
*W4*	chr3:79549780G>A	missense	All White
*W5*	chr3:79545900delC	small deletion	Sabino-like
*W6*	chr3:79573754C>T	missense	Sabino-like to All White
*W7*	chr3:79580000C>G	splice site	All White
*W8*	chr3:79545374C>T	splice site	Sabino-like
*W9*	chr3:79549797C>T	missense	All White
*W10*	chr3:79566925_79566928del	small deletion	Sabino-like to All White
*W11*	chr3:79540429C>A	splice site	All White
*W12*	chr3:79579755_79579779delAGACG	small deletion	Sabino-like
*W13*	chr3:79544066C>G	splice site	All White
*W14*	chr3:79544151_79544204del	gross deletion	All White
*W15*	chr3:79550351A>G	missense	Sabino-like to All White
*W16*	chr3:79540741T>A	missense	All White
*W17a*	chr3:79548265T>A	missense	All White
*W17b*	chr3:79548244A>G	missense	All White
*W18*	chr3:79553751C>T	splice site	Sabino-like
*W19*	chr3:79553776T>C	missense	Sabino-like
*W20*	chr3:7948220T>C	missense	No markings to Sabino-like
*W21*	chr3:79544174delG	small deletion	Sabino-like
*W22*	chr3:79548925_79550822del1898	gross deletion	Sabino-like
*W23*	chr3:79578484C>G	splice site	All White
*W24*	chr3:79545245C>T	splice site	All White
*W25*	chr3:77769878T>C	missense	All White
*W26*	chr3:79544150del	small deletion	Sabino-like
*W27*	chr3:79552028A>C	missense	All White
*W28*	chr3:79579925_79581197del	gross deletion	Sabino-like
*W30*	chr3:79548244T>A	missense	All White
*W31*	chr3:79618532_79618533insT	fs nonsense	Sabino-like
*W32*	chr3:79538738C>T	missense	No markings to Sabino-like
*W33*	chr3:79545248T>A	missense	Sabino-like
*W34*	chr3:79566881T>C	missense	No markings to Sabino-like
*W35*	chr3:79618649A>C	UTR variant	No markings to Sabino-like
*SB1*	ch3:9544206A>T	splice site	Sabino to All White

**Table 2 animals-14-00517-t002:** Allele frequencies and average white scores (AWSs) where computable for 36 *KIT* alleles in a sample of 8334 horses.

Allele	Frequency	AWS	Allele	Frequency	AWS	Allele	Frequency	AWS
*W1*	0	-	*W13*	1.20 × 10^−4^	-	*W24*	0	-
*W2*	0	-	*W14*	0	-	*W25*	0	-
*W3*	1.80 × 10^−4^	34	*W15*	6.60 × 10^−4^	23.6	*W26*	0	-
*W4*	0	-	*W16*	0	-	*W27*	0	-
*W5*	2.40 × 10^−4^	29	*W17a*	0	-	*W28*	0	-
*W6*	0	-	*W17b*	0	-	*W30*	1.20 × 10^−4^	34
*W7*	0	-	*W18*	0	-	*W31*	6.00 × 10^−4^	28.66
*W8*	0	-	*W19*	1.62 × 10^−3^	24.5	*W32*	6.08 × 10^−2^	2.63
*W9*	0	-	*W20*	2.05 × 10^−1^	9.54	*W33*	0	-
*W10*	0	-	*W21*	0	-	*W34*	1.54 × 10^−2^	10.34
*W11*	0	-	*W22* ^a^	6.00 × 10^−4^	19.5	*W35*	1.68 × 10^−1^	10.7
*W12*	0	-	*W23*	0	-	*SB1*	1.40 × 10^−2^	34

^a^—*W22* is thought to have occurred on the *W20* background [8,15]. Since all observed *W22* horses have at least one copy of *W20*, the white score for the *W22* allele represents horses with only one copy of *W22* and one copy of *W20*.

**Table 3 animals-14-00517-t003:** Haplotypes and frequency (*F*) of *KIT* variants identified in a dataset of 8334 horses using BEAGLE 5.4 phasing software.

Haplotype	*n*	*F*	Haplotype	*n*	*F*	Haplotype	*n*	*F*
*WT*	10,105	6.06 × 10^−1^	*W34*	8	4.80 × 10^−4^	*W20W32*	49	2.94 × 10^−3^
*W3*	1	6.00 × 10^−5^	*W35*	1905	1.14 × 10^−1^	*W32W35*	554	3.32 × 10^−2^
*W5*	2	1.20 × 10^−4^	*SB1*	6	3.60 × 10^−4^	*W32SB1*	215	1.29 × 10^−2^
*W13*	1	6.00 × 10^−5^	*W5W32*	2	1.20 × 10^−4^	*W34W35*	223	1.34 × 10^−2^
*W15*	11	6.60 × 10^−4^	*W13W35*	1	6.00 × 10^−5^	*W19W34W35*	23	1.38 × 10^−3^
*W19*	4	2.40 × 10^−4^	*W20W35*	92	5.52 × 10^−3^	*W3W34W35*	2	1.20 × 10^−4^
*W20*	3246	1.95 × 10^−1^	*W20SB1*	2	1.20 × 10^−4^	*W20W32W35*	5	3.00 × 10^−4^
*W30*	2	1.20 × 10^−4^	*W20W22*	10	6.00 × 10^−4^	*W20W34W35*	1	6.00 × 10^−5^
*W31*	9	5.40 × 10^−4^	*W20W31*	1	6.00 × 10^−5^	*SB1W20W32*	11	6.60 × 10^−4^
*W32*	177	1.06 × 10^−2^						

**Table 4 animals-14-00517-t004:** Fourteen genotypes with at least 3 *KIT* variants and associations with white spotting phenotypes using ANOVA.

Variants	*n*	*p*	AWS	More White than…
*SB1W32/W20*	20	2.2204 × 10^−16^	29.50	*W20/X*, *W32/X*
*W32W35/W35*	23	1.5416 × 10^−4^	12.30	*W32/X*, *W35/X*
*SB1W32/W35*	6	1.8054 × 10^−11^	26.66	*W32/X*, *W35/X*
*SB1W32/W32W35*	3	1.4990 × 10^−9^	32.66	*W32/X*, *W35/X*
*W32W35/W20*	39	1.9670 × 10^−5^	14.71	*W20/n*, *W32/X*
*W19W34W35/n*	15	1.4504 × 10^−5^	23.26	*W34/n*, *W35/n*
*W32W35/W32W35*	15	2.0794 × 10^−3^	9.46	*W32/X*
*W32W35/W32*	3	0.2032	4.66	-
*W34W35/W20*	4	0.3165	13.00	-
*W20W35/W20*	3	0.0642	18.33	-
*W34W35/W34*	15	0.3295	8.46	-
*W34W35/W34W35*	3	0.2234	13.33	-

## Data Availability

All BEAGLE phased genotypes are available in Appendix A.

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
