# Peer review of "Population Analysis Identifies 15 Multi-Variant Dominant White Haplotypes in Horses"

_animals, 2024, doi:10.3390/ani14030517_

Round 1

Reviewer 1 Report

Comments and Suggestions for Authors

The paper details the haplotyping of mutations of the KIT gene, which is responsible for white patterning in coat colour variation. The authors describe new haplotypes with multiple KIT mutations by use of Beagle informatics tools and compare the results to phenotyping data derived from photos. The authors also provide recommendations on the presentation of haplotypes to more accurately present the variants present.

Although this paper contains a lot of interesting data and analysis, there are some serious editing issues with this manuscript that need to be addressed before publication. Something either went wrong during the submission process or it wasn't proof-read beforehand.  There are still instructions to authors embedded in the text (eg line 110, 344, 363) and the references are not in order of citation, but in alphabetical order.

Please find specific questions and comments below::

Line 86. Was permission sought for analysing data from all 8334 horses, or just those where photographs were provided?

Line 96. Please provide some additional details on the genotyping platform for readers who may not have access to the original paper.

Line 102. No need to add all the references again for the sequencing.

Table 1. References not in the same format as the rest of the paper. These will need to be numbered

Table 2. This table could do with re-formatting for ease of use.

Table 2. W13 does not have an AWS associated with it, but later in text in line 176 it's described as having a score greater than 20.

Table 2. Which other alleles was W22b in complete linkage with? (Later in the text this is shown to be W20). How were the AWS scores calculated for these linked alleles?

Line 143. How was the estimated population size calculated? Was this determined by Beagle 5.4?

Table 3. This table could do with re-formatting for ease of use.

Table 3. Should 'frequency' be 'estimated frequency' if it's based on the estimated population size?

In Table 4 and related results, the authors compare the average white score in multi-variant haplotypes on the photographs available. As the scoring system is on multiple areas (shown on Supplementary Figure 1), has the effect of the multi-variant haplotypes on specific areas of the horse been explored, or would a much larger data set be required to distinguish any differences from genetic vs random stochastic events as discussed in the introduction?

How do Etalon Inc currently report KIT alleles?

Author Response

Thank you for taking the time to review our manuscript. We appreciate your comments and thoroughness. In response to your comments we made the following changes: 

  1. We removed all the errors caused by the submission process and reordered our citations in order of appearance 
  2. Added that we received consent to analyze all data, and for the photos we used 
  3. Added details on our genotyping and sequencing platform/methodology, including a citation to a best-practice GATK pipeline 
  4. Removed the duplicate reference 
  5. Fixed the references in the table to be numbered 
  6. Added shading to differentiate the vertical blocks
  7. Deleted the reference to W13 AWS as we could not calculate an average 
  8. Changed the footer to table 2 to specifically mention W22 and the linked W20 as well as how we treated the AWS for that variant
  9. Mentioned that the estimated population size was calculated by BEAGLE
  10. Reformatted Table 3 the same way as table 2 (see 8)
  11. Changed frequency to F, mentioned in the methods that frequencies are always reported with respect to our sample of 8334 horses, VCFtools –freq does not use the BEAGLE population size for its calculations 
  12. We attempted to perform this analysis, but there weren't enough photos to validate those claims due to the sheer amount of white spotting alleles. We are starting to find that alleles in other genes are contributing more to differently located white markings, and that manuscript is in review. As far as KIT alleles go, there has been no full analysis on the heritability of locations of markings other than the face and limbs together or all white phenotypes. We address these limitations and more in an added paragraph in the discussion 
  13. Etalon is currently adapting to the guidelines set forth by this publication. Etalon currently reports variants as they unphased recommendation (eg. W19/n W34/n W35/W35). Etalon has web pages dedicated to each KIT mutations detailing the observed phenotypes, and states if the mutation has been observed in the homozygous state.

Reviewer 2 Report

Comments and Suggestions for Authors

Color genetics is of interest within the horse industry and is a relatively new science requiring further work to support advancement of breeding sciences within the industry. As such, the authors are commended on the work they have presented within the submitted manuscript. Minor additional revisions, however, are recommended for further improvement.

To begin with, authors should include a clear objective statement within both the summary and the abstract. Within the introduction, while an objective statement is given at the end of the introduction, authors should remove after the objective statement the sentences concerning results, lines 85-88, and move this information to the results section, and then, add in a clear hypothesis statement after the given objective statement. Also, for the introduction, divide the last paragraph into two paragraphs, dividing at line 74 so that the first sentence for the newly created last paragraph begins with "Although". This newly created last paragraph will assist with emphasizing the objectives of the study. Conclude this last paragraph with not only the objective and hypothesis statements, but also a statement on the value of the research.

For the methods section, make sure to begin with clearly indicating whether institutional review boards were utilized for review of research protocol and ethical care was taken in research steps associated with interaction with animals and their owners. Include information about receiving owner consent. Provide further information concerning breed type utilized for photo grading and give specifics on inclusion/exclusion criteria on photos since the sample size is much smaller than that utilized for the rest of the study. Include a figure that shows examples of the photos utilized and/or not utilized to illustrate the inclusion/exclusion criteria. Further, give details on qualifications for the single, blinded observer mentioned in line 118. In addition, within the methods section, add in the paragraph before Table 2, lines 129-136, reference of the table and what it represents as currently the table is not referenced until line 150.

As for the discussion section, add in citations to references to support points given throughout the discussion as it appears at times more opinion based without the use of references. Ensure references utilized reflect more up to date research as currently 1/3 of the references cited were published 2010 or later. Genetics is a relatively new science as it relates to horses so that research should be based within the last decade. For figure 1, lines 301-302 should be moved to acknowledgements given in lines 357-358. Further, indicate within lines 296-301 whether these photos within figure 1 were utilized for the grading system for this study. Finally, end the discussion section before going into the conclusions with a paragraph on study limitations. This should cover, but not limited to whether sample size was adequate for sampling all breeds, were all breeds equally sampled, would in person grading of horse phenotype be more reliable than photos, and would use of a control assist in verification of data? How were limitations addressed? What suggested improvements in research methodology could assist future studies? 

Comments on the Quality of English Language

Thoroughly review over the manuscript as there are several typos. Below are a few areas to correct:

Line 88: Add in a space before the title for methods.

Line 109: Add a period before "We".

Line 110-111: Remove "3." and "Results" (appears to be a misplaced heading).

Lines 112-114: Remove these lines as it appears to be comments to the authors, maybe reminders for the authors that did not get removed in the final draft.

Line 186: Bold "Table 4".

Lines 283-290: Adjust font to match the font utilized throughout the manuscript.

For the methods, results, and discussion sections, "Single Group" and "Single group" were utilized, thus, to stay consistent, select either to capitalize or not "group". 

Lines 344-353: Delete these lines as it appears this information was pasted from author's guidelines given by the journal. Within this section, give clear details including protocol approval numbers for institutional review of research protocol. 

Lines 363-368: Delete lines as it appears this information was pasted from author's guidelines given by the journal. 

For referencing, correct so that numbers for references are given within the brackets only as currently brackets are utilized for most reference numbers but also additional numbers are given in front of the brackets.

Author Response

Thank you for taking the time to review our manuscript. We appreciate your comments and thoroughness. In response to your comments we made the following changes: 

  1. Introduction and Front Matter: 
    1. We added explicit objective statements to both the simple summary and abstract 
    2. Removed the results presented in the introduction paragraph
    3. Added a hypothesis to the introduction 
    4. Divided the last paragraph of the introduction to emphasize our objectives 
    5. Added to our value of research statement in the introduction
  2. Methods 
    1. Expanded on the consent of horse owners and the protocols used by our study
    2. Added a citation for the genotyping platform (best practice GATK pipeline) we used and elaborated on the sequencing performed 
    3. Added the guidelines we followed for our research on animals 
    4. Expanded on our Board Review statement in the back matter 
    5. Added a figure depicting inclusion/exclusion criteria for photographs and an explanation for exclusion.
    6. Added to the qualifications of the single blinded observer 
    7. We moved the reference to table 2 earlier, but did not reference it in the methods section as it contains results that haven’t been described. 
  3. Discussion
    1. Added numerous citations throughout the discussion to support our claims through previous research 
    2. Moved the “Thank you” to the acknowledgements
    3. Indicated that the photos in the figure were used for phenotyping 
    4. Added a paragraph to the end of the discussion section detailing the limitations of our study including: our phenotyping strategy, breed bias, sample bias, and how future research could adapt our strategy. 
  4. English Editing 
    1. Added a new line spacer between end of paragraph and the Methods section
    2. Added a period before the We
    3. Removed the 3.Results misplaced heading
    4. Removed the comments to authors 
    5. Bolded table 4
    6. Changed the font to match throughout the manuscript 
    7. Made all occurrences of “group” match for consistency 
    8. Deleted the instructions for authors, and rewrote the section as appropriate
    9. Removed a suppl. File that was not meant for this manuscript
    10. Fixed references order and ensured the numbers do not appear twice

Round 2

Reviewer 1 Report

Comments and Suggestions for Authors

The paper is much improved and the comments from the previous version have been addressed.